# Mechanical Equilibrium Dynamics Controlling Wetting State Transition at Low-Temperature Superhydrophobic Array-Microstructure Surfaces

**Yizhou Shen [1,\*], Xinyu Xie [1], Jie Tao [1,2], Haifeng Chen [3], Zeyu Cai [1], Senyun Liu [4] and Jiawei Jiang [1]**

[1] College of Materials Science and Technology, Nanjing University of Aeronautics and Astronautics, Nanjing 210016, China; xiexinyu@nuaa.edu.cn (X.X.); taojie@nuaa.edu.cn (J.T.); caizeyu@nuaa.edu.cn (Z.C.); jiangjiawei@nuaa.edu.cn (J.J.)

[2] Jiangsu Collaborative Innovation Center for Advanced Inorganic Function Composites, Nanjing Tech University, Nanjing 210009, China

[3] Department of Materials Chemistry, Qiuzhen School, Huzhou University, 759# East 2nd Road, Huzhou 313000, China; headder@zjhu.edu.cn

[4] Key Laboratory of Icing and Anti/De-Icing, China Aerodynamics Research and Development Center, Mianyang 621000, China; liusenyun@cardc.cn

\* Correspondence: shenyizhou@nuaa.edu.cn

**Abstract:** Superhydrophobic materials are significant for engineering applications in the anti-icing field because of their non-wetting property. The interface physical mechanisms of non-wetting properties are important to promote real applications of superhydrophobic surfaces, especially under low-temperature conditions. Here, we found that low temperature could induce the wetting state transition from a Cassie–Baxter state to a Wenzel state. This transition occurred at 14 °C (and 2 °C) on superhydrophobic surfaces with pillar heights of 250 μm (and 300 μm). As a consequence, the driving-force of the Cassie-Wenzel (C-W) wetting transition was induced by the contraction of air pockets on cooling, and the pressure of air pockets supporting the droplet decreased with the contraction degree. Decreasing the pressure of air pockets broke the mechanical equilibrium at the solid–liquid contact interface, and the continuous contraction overcame the resistance in the C-W wetting transition. Based on the analysis of work against resistance in the C-W wetting transition, lower C-W wetting transition temperature was mainly attributed to a higher pillar, which produced more work against resistance to require more energy. This energy was directly reflected by the energy required for continuous contraction of air pockets. Superhydrophobic surfaces with higher pillar structure remain stable non-wetting property at low-temperature conditions. This work provides theoretical support for the application of superhydrophobic materials in low-temperature environments.

**Keywords:** wetting state transition; non-wetting behavior; low temperature

## 1. Introduction

Wettability is one of the important phenomena of solid surfaces [1–3]. The superhydrophobic phenomena have aroused many concerns because of their great importance for a range of applications, especially in the field of anti-icing to remove supercooled droplets [4,5]. Superhydrophobicity is usually achieved on micro-nanostructure surfaces with a certain condition of chemical compositions [6]. Theoretically, this remarkable property is due to the air pockets being trapped by the microscopic structures on superhydrophobic surfaces, leading to the Cassie-Baxter wetting state. The state shows a composite contact interface of mixed solid-liquid and liquid-air interfaces. Therefore, it results in a smaller contact area between a solid and liquid and higher non-wetting property. However, the Cassie-Baxter wetting state is not always a thermodynamically stable state, and a transition can occur to the more stable Wenzel wetting state. External factors (pressure,

vibration, temperature, etc.) can induce the occurrence of the Cassie-Wenzel (C-W) wetting transition [7–10]. For example, Fu et al. reported that the wetting state depends on substrate temperature, and the wetting state of the droplet tends to change as temperature decreases [11]. This indicates that it is challengeable to sustain the Cassie-Baxter wetting state for a droplet on the superhydrophobic materials in low-temperature conditions. In other words, it is disadvantageous for the application of superhydrophobic materials in anti-icing at low temperatures. Therefore, the study of wetting state transition driven by low temperature is significant for the optimization of superhydrophobicity and the application of superhydrophobic materials.

Considerable efforts have been directed toward understanding the C-W wetting transition process and the underlying interface mechanism [12,13]. Researchers have often judged the C-W wetting transition according to the change in contact angle (CA), and explain it via the mechanical equilibrium [14,15]. Gerber et al. compared the dynamic wetting behavior and revealed the difference in the C-W wetting transition via the value of the CA on different substrates, and found that the CA parameter plays an important role in the analysis of the C-W wetting transition [16]. Papadopoulos et al. reported that the C-W wetting transition is an impalement of a droplet into the surface structure, and monitored the evolution of air pockets beneath the droplet [17]. Fang et al. also investigated the wetting transition on superhydrophobic surfaces, and proposed that the pressure of air pockets provides support for a droplet in the Cassie-Baxter wetting state [18]. Based on previous research, we think that the stable air pockets are necessary for maintaining the Cassie-Baxter wetting state of a droplet, and it affects the C-W wetting transition to a great extent.

Herein, temperature was proved to induce the wetting state transition, where the droplet tends to be in a Cassie-Baxter wetting state on a higher-temperature surface. In addition, lower-temperature conditions provide a certain driving force to induce the occurrence of the C-W wetting transition. To investigate the C-W wetting transition mechanism in detail, we designed and prepared three types of superhydrophobic surfaces with pillar array-microstructures through photolithography and spraying technologies. On these sample surfaces, the C-W wetting transition process of a 10 μL droplet was recorded by a CCD (Charge Coupled Device) camera, as the sample surface was cooled from room temperature to zero degrees Celsius. The apparent change in the solid-liquid interface state occurred instantaneously, which was well-reflected by the CA parameter. On this basis, an instantaneous balance was revealed around both aspects of resistance and driving force induced by a low-temperature condition.

## 2. Materials and Methods

As shown in Figure 1, the surfaces covered with regular cylinder array structures were prepared by photolithography on Ti6Al4V (composition (wt.%): $\leq 0.3\%$ Fe, $\leq 0.1\%$ C, $\leq 0.05\%$ N, $\leq 0.015\%$ H, $\leq 0.2\%$ O, 5.5–6.8% Al, 3.5–4.5% V, and the rest was Ti) substrates. Firstly, the substrate was thoroughly cleaned ultrasonically with acetone, ethanol, and distilled water for 30 min, and then dried immediately under the cold wind condition. Subsequently, the substrate with photoresist was processed by photolithography to form cylinder structures. The diameter of pillar structure was the same as the diameter of the pattern on the mask, and the height was determined by the thickness of the photoresist. In the experiment, the time of UV light exposure was changed to match different thicknesses of photoresist. The above-surface was sprayed with modified hydrophobic $SiO_2$ nanoparticles to obtain superhydrophobicity (the detailed prepared procedures are provided in the Section 1 of Supplementary Material). In order to obtain the superhydrophobic array-microstructure surface, 0.05 g of hydrophobic $SiO_2$ nanoparticles was ultrasonically dispersed into 10 mL of ethanol solution for 30 min. Following this, the mixed system was sprayed on the substrate by means of an airbrush at 0.4 MPa.

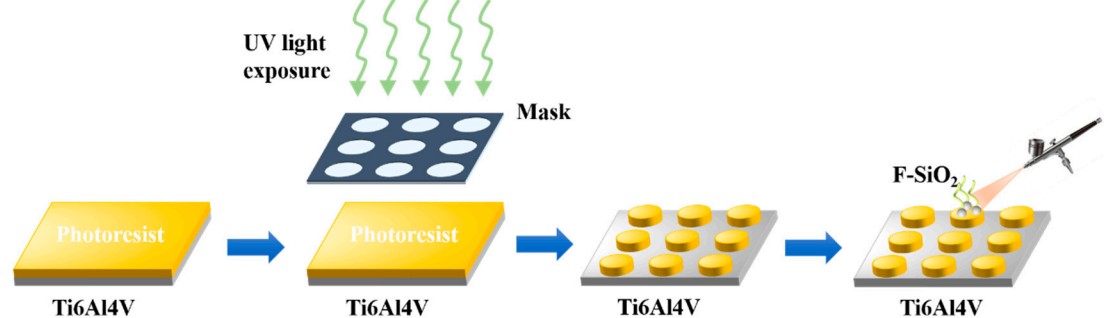

**Figure 1.** Schematic diagram of the procedure for the preparation of superhydrophobic array-microstructure surface.

Figure 2a–c shows the three-dimensional topographies of microstructures with regular pillar arrays constructed using the photolithography method. It can be clearly seen that the surfaces are evenly covered by array-microstructures with same size of pillar diameter and center distance, where the diameter D is 500 μm and the center distance L is 850 μm. The pillar array structures are coated by a layer of $SiO_2$ nanoparticles to produce the nanostructures for the final superhydrophobicity (see Figure 2d–f). Additionally, the final sample surfaces with the pillar height from 200 to 250 μm and 300 μm are labelled as H1, H2, and H3, respectively. Also, the chemical composition of surface is illustrated in Figure S1. On this basis, a 10 μL droplet can nearly suspend on the superhydrophobic array microstructure surface, and the surfaces maintain high superhydrophobicity with a CA value of 152° ± 2° at a room temperature of 25 ± 0.5 °C and a relative humidity of 65% ± 5% (see Figure 2d–f). As a consequence, the apparent solid-liquid contact interface is composed of solid-liquid and liquid-air interfaces, which well-conforms to the typical Cassie-Baxter wetting model.

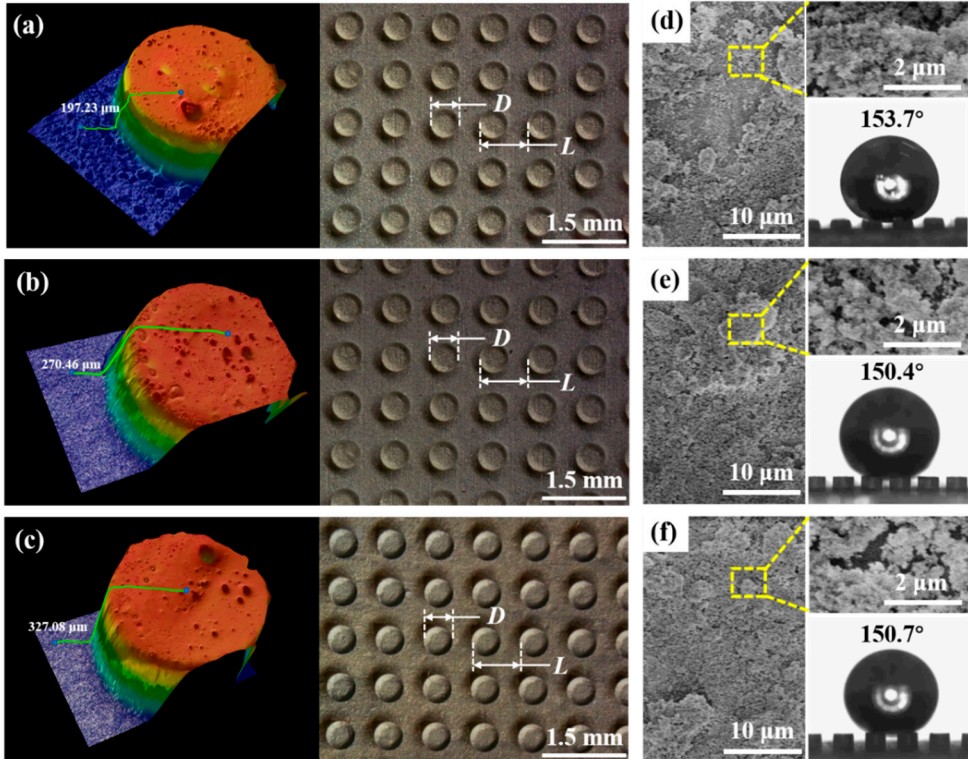

**Figure 2.** Morphologies of the superhydrophobic micro-nanostructure surfaces. (**a–c**) Three-dimensional topographies of array microstructures. (**d–f**) Higher magnification SEM images of the nanostructures on the surfaces of the array microstructures.

## 3. Results

### 3.1. Non-Wetting Behavior at Low Temperature

From the mechanical perspective, the balance relation at the solid-liquid contact interface can be obtained to discuss the superhydrophobicity of the array micro-nanostructure superhydrophobic surface, as shown in Figure 3a. It can be easily concluded that the superhydrophobicity is a result of the pressure of air pockets and braced force to resist the gravity of a water droplet. The detail is that the pressure of air pockets supports the droplet across the liquid-air interface. As the solid-liquid contact interface being sensitive to the temperature, we designed an experiment to reduce the surface temperature to 0 °C, and analyzed the low-temperature non-wetting behaviors. The detailed experiment procedures are provided in the Section 3 in the Supplementary Material (see Figure S2). It was also discovered that the low-temperature non-wetting property is more important to anti-icing applications of superhydrophobic surfaces.

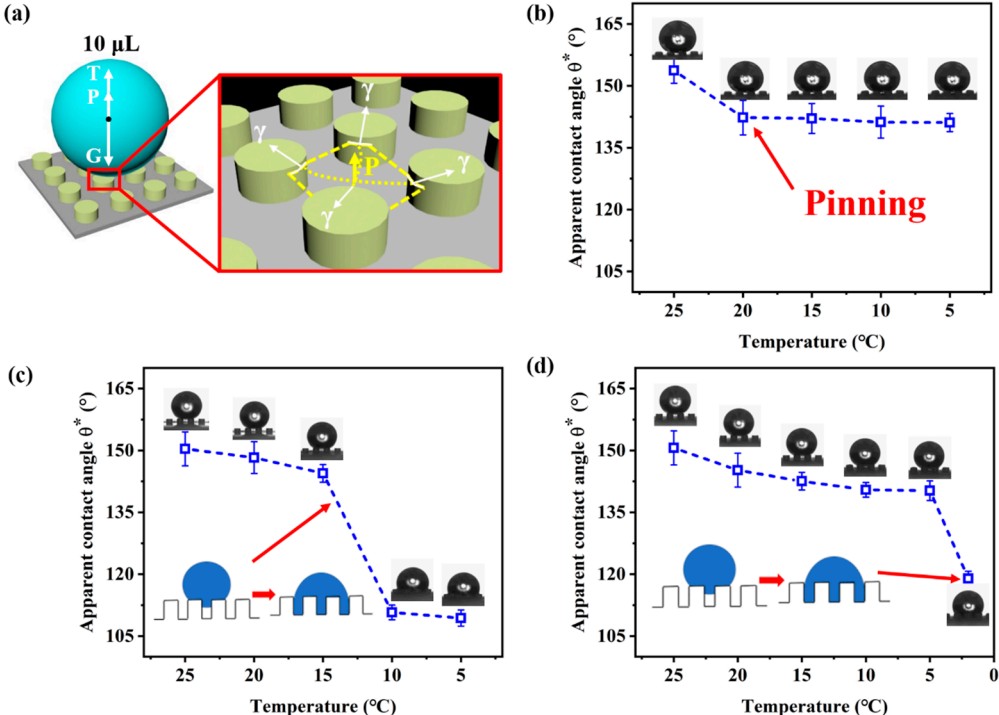

**Figure 3.** (**a**) Schematic diagram of wetting state at room temperature. (**b–d**) Changes in contact angle with temperature on the superhydrophobic surfaces with the pillar height from 200 to 250 μm and 300 μm.

The wetting state transition affected by low-temperature is valuable for studying the low-temperature non-wetting property. The non-wetting behavior of droplets on the surface is an important aspect of wetting state transition, which is mainly evaluated via the CA parameter. Normally, the CA exhibits an obvious decreasing tendency when the wetting states change from Cassie-Baxter state to Wenzel state. In addition, the liquid penetrates into the structure spaces and replaces the trapped air pockets below the droplet. Therefore, to study the dynamic wetting transition of droplets on the superhydrophobic array-microstructure surfaces, we analyzed the dynamic change in CA. Figure 3b–d shows the wetting state and CA of a droplet on the surfaces under the cooling condition. The results not only revealed that the low-temperature could induce a decrease in CA, but also that the structures would affect the decreasing trend in CA ($\theta^*$). In contrast with the H2 and H3 surfaces, the $\theta^*$ on H1 was basically unchanged and remained around 141.0° ± 2°, which meant that no C-W wetting transition occurred in this situation, as depicted in Figure 3b. However, in Figure 3c, the CA on the H2 surface gradually decreased from 150.4° to 109.3° under the effect of reducing temperature. The curve also indicates that

there was an instantaneous transition point of the decreasing CA. The CA decreased by $35° ± 2°$ at this point, which meant that the droplet collapsed into the pillar structures to become a Wenzel wetting state. The apparent solid-liquid interface state occurred instantaneously during the change in this situation, resulting in an occurrence of the C-W wetting transition. According to the temperature curve (details are provided in the Section 3 of the Supplementary Material, Figure S3), the C-W wetting transition temperature was around 14 °C. Meanwhile, the C-W wetting transition also occurred on the H3 surface. Figure 3d illustrates that the CA decreased from 150.7° to 118.9°, and the C-W wetting transition temperature during this process was close to 2 °C. Compared with the H2 surface, it is obvious that the droplet was suspended on the structure in a Cassie-Baxter wetting state for a longer period, and the C-W wetting transition temperature differed by 12 °C.

### 3.2. State of the Solid-Liquid Contact Interface at Low Temperature

The contact angle directly reflects whether the droplet converted from a Cassie-Baxter wetting state to a Wenzel wetting state under low-temperature conditions. On the basis of the dynamic transition shown in the Section 5 of the Supplementary Material (Video S1, Video S2 and Video S3), we found that the state of the solid-liquid contact interface changed significantly during this process, including solid-liquid and liquid-air interfaces. In order to further reveal the change phenomenon of the solid-liquid contact interface state, we analyzed the contact interface state and established a state model under the low-temperature action according to the dynamic transition. As shown in Figure 4a, the model can be divided into two courses called state I and state II, including the spreading of the solid-liquid interface on the upper surface of the pillar structure and the penetration of the liquid-air interface into the structure gap. Regarding state I, the solid-liquid interface spread under the induction of a low temperature, and it was pinned at the edge of the upper surface of the pillar structure. The interface tension at point A was in a mechanical equilibrium state. As illustrated in Figure 4b, the solid-air interface tension was vertical, which promoted the balance of interface tensions. After that, if the driving force induced by low temperature was sufficiently large, the liquid-air interface in the non-equilibrium state penetrated into the structure gap. State II represents a process in which the liquid-air interface only penetrated into the structure gap to maintain balance. Finally, the liquid-air interface touched the bottom of the substrate and the state of the droplet instantly changed to the Wenzel wetting state. Here, Figure 4c illustrates that the solid-liquid contact state at 25 °C conformed to the Cassie-Baxter wetting model with a composite contact interface. However, in a clear difference from the Cassie-Baxter wetting model on the H1 surface at 0 °C, the solid-liquid contact state on the H2 and H3 surfaces conformed to the Wenzel wetting model. Moreover, compared with the contact interface state of the C-W wetting transition point on these surfaces as shown in Figure 4c, the liquid-air interfaces on the H2 and H3 surfaces touched the pillar bottom. The results showed that the solid-liquid contact interface state of a droplet on the H2 and H3 surfaces included state I and state II before the C-W wetting transition, while it only included state I on the H1 surface. Considering the solid-liquid contact interface as being sensitive to the temperature, the equilibrium mechanism at the solid-liquid contact interface can be obtained to discuss the non-wetting property at a low temperature. Based on analysis of the contact interface state, it is obvious that the low-temperature condition provides a driving force to induce the penetration of liquid-air interface into the structure gap, leading to the occurrence of the C-W wetting transition. Thus, the equilibrium mechanism at the liquid-air interface is considered to be an important factor in revealing the C-W wetting transition mechanism.

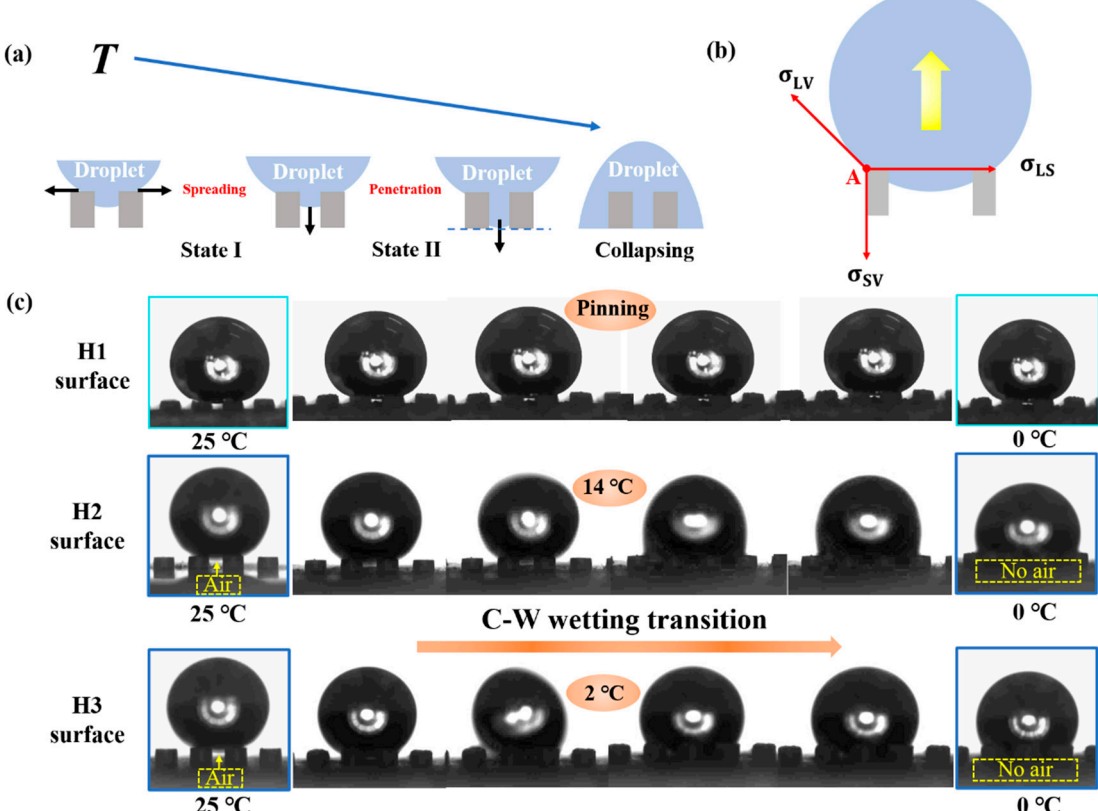

**Figure 4.** (**a**) Schematic diagram of contact line movement. (**b**) Schematic diagram of the direction of interface tensions when the contact line extends to the edge. (**c**) Dynamic C-W wetting transition.

## 4. Discussion

Low temperature can provide a driving force to induce the C-W wetting transition, which is directly reflected by the decrease in CA. Moreover, the results indicate the stability of the non-wetting property of superhydrophobic surfaces with a pillar height ranging from 200 to 250 μm and 300 μm under low-temperature conditions. According to the state model of the solid-liquid contact interface, maintaining the stability of the liquid-air interface can effectively keep the non-wetting property of superhydrophobic surfaces. Furthermore, the equilibrium mechanism at the liquid-air interface is an important factor revealing the C-W wetting transition mechanism. Theoretically, the mechanical equilibrium across the liquid-air interface is a pressure equilibrium, which is interpreted as the pressure difference between the pressure of the droplet and the pressure of air pockets. When the pressure difference exceeds the critical pressure difference, the interface no longer stays pinned and the C-W wetting transition may also occur. The pressure of air pockets provides support for a droplet to reduce the pressure difference. Thus, the pressure stability of air pockets trapped by the structure on the superhydrophobic surface determines the stability of the non-wetting property, where the droplet keeps a more durable Cassie-Baxter wetting state under the support of the stable pressure of air pockets. As shown in Figure 5a, the pressure of air pockets ($P_{air}$) is a supporting force across the liquid-air interface resisting the pressure of a droplet ($P_{water}$) generated by liquid-air interface tension. When the pressure difference ($P_{water} - P_{air}$) exceeds the critical pressure difference, the interface in a non-equilibrium state penetrates into the structure gap. In accordance with the mechanical mechanism at the interface, the C-W wetting transition can be inferred as the result of the pressure change of air pockets caused by the low temperature. It is well-known that the physical property of air affected by temperature is expansion on heating and contraction on cooling. Therefore, with a decrease in temperature, the pressure of air pockets with contraction decreases to induce the penetration of the liquid-air interface, resulting in the occurrence

of the C-W wetting transition. In addition, in the process of contraction on cooling, the pressure of air pockets is proportional to the product of the volume of air pockets ($V$) and the temperature ($T$), expressed as $P_{air} \propto VT$. By comparatively analyzing the volume of air pockets beneath the droplets on the H1, H2, and H3 surfaces, we observed that the volume of air pockets on these surfaces was proportional to the height of the pillar structure, which is expressed as $V_1$, $V_2$, and $V_3$. Thus, in terms of pillar height, this relationship is expressed as $V_1 < V_2 < V_3$. Obviously, due to the insufficient volume of air pockets on the H1 surface, the pressure of air pockets that decreased via contraction was insufficient, and the pressure difference could not exceed the critical pressure difference to induce the C-W wetting transition at a low-temperature condition. Correspondingly, under the action of cooling, the pressure difference exceeded the critical pressure difference on the H2 and H3 surfaces to induce the occurrence of the C-W wetting transition. Meanwhile, the continuous contraction of air pockets on cooling in the process of the C-W wetting transition was observed to overcome resistance hindering the penetration of the interface, as shown in the schematic in Figure 5b. The resistance was actually equal to the critical pressure difference. From the viewpoint of resistance, the work performed against resistance in the C-W wetting transition is thus equal to $P_C \Delta H$, the $P_C$ is the critical pressure difference and it is a constant independent of height, and $\Delta H$ is the penetration depth of the liquid-air interface [9]. It is easy to understand that penetration depth is a parameter determined by the pillar height, where a higher pillar produces deeper penetration depth. Therefore, the work performed against resistance is conclusively determined by the height of the pillar structure in the C-W wetting transition; a higher pillar will result in more work against the resistance. As analyzed above, we conclude that the C-W wetting transition on a superhydrophobic surface with a higher pillar needs more energy to overcome the work against resistance. This energy is obtained through the work by the pressure of the droplet, which is induced by the continuous contraction of air pockets on cooling. The energy obtained by the droplet in the C-W wetting transition is equivalent to the energy required for contraction of air pockets. Thus, the C-W wetting transition on the H3 surface needed more energy to overcome the work done against resistance, showing a lower C-W wetting transition temperature. Compared with the C-W wetting transition temperature of 14 °C on the H2 surface, the C-W wetting transition temperature on the H3 surface was 2 °C, because a higher pillar produces more work against resistance for a droplet to convert to a Wenzel wetting state.

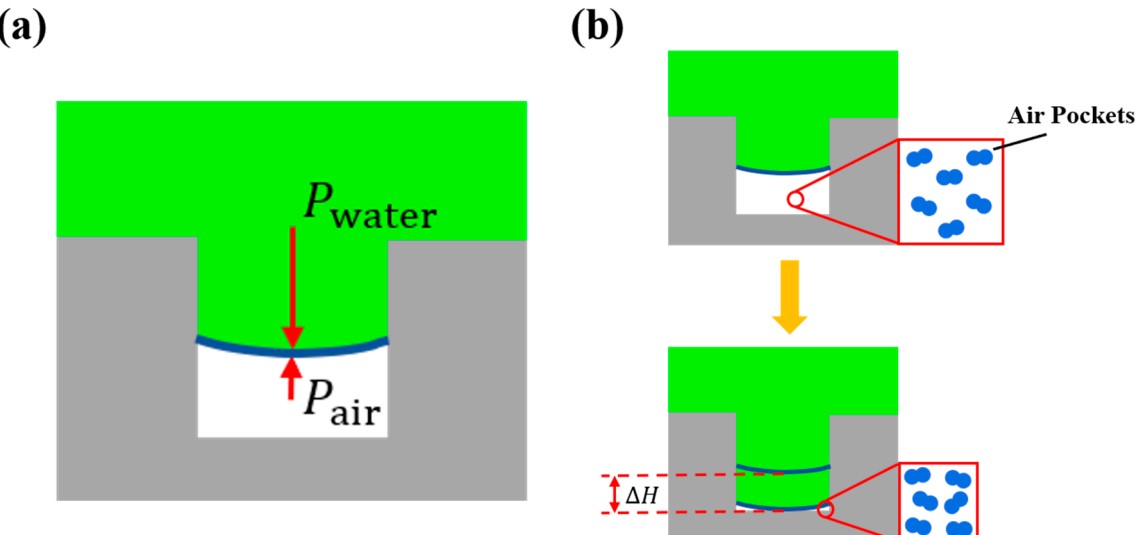

**Figure 5.** (**a**) Schematic diagram of the force at the liquid-air interface. (**b**) Schematic diagram of continuous contraction of air pockets on cooling to overcome resistance in the C-W wetting transition.

In order to further verify that the difference in the C-W wetting transition temperature results from the different work done against resistance, we first quantitatively compared the work done against resistance and then quantitatively analyzed the energy obtained in the C-W wetting transition for the H2 and H3 surfaces. According to the analysis of work against resistance, the penetration depth on the H2 and H3 surfaces reflected the relationship of work against resistance. The detailed analysis is provided in the Section 4 in the Supplementary Material. The work performed against resistance on the H2 and H3 surfaces is expressed as $W_{H2}$ and $W_{H3}$; the relationship is

$$W_{H3} \approx 2W_{H2} \tag{1}$$

In addition, the energy obtained in the C-W wetting transition is equivalent to energy required for continuous contraction of air pockets on cooling. The energy ($W_{air}$) required for continuous contraction of air pockets is expressed as $W_{air} \propto VT$. In terms of the C-W wetting transition temperature, the energy required for continuous contraction of air pockets on the H2 and H3 surfaces is expressed as $W_{airH2}$ and $W_{airH3}$; the relationship is

$$W_{airH3} \approx 2W_{airH2} \tag{2}$$

The work done against resistance on the H3 surface was twice as much as that on the H2 surface. Similarly, the energy obtained in the C-W wetting transition on the H3 surface was twice as much as that on the H2 surface. The results demonstrate that the continuous contraction of air pockets on cooling was mainly to overcome the resistance, and the lower C-W wetting transition temperature on the H3 surface was attributed to the higher pillar, which produced more work against resistance for a droplet to convert to a Wenzel wetting state. A lower C-W wetting transition temperature means more energy required for contraction of air pockets, which reflects that more energy can be obtained during the C-W wetting transition.

## 5. Conclusions

In summary, it was essential to analyze the mechanism of the wetting state transition on low-temperature superhydrophobic array-microstructure surfaces to develop a fundamental theory. Low-temperature conditions can provide a certain driving force to induce the occurrence of the C-W wetting transition. The driving force of the C-W wetting transition is the contraction of air pockets on cooling, resulting in the decrease in pressure of air pockets to break the mechanical balance at the solid-liquid contact interface. Moreover, the continuous contraction of air pockets on cooling mainly overcomes the resistance. Lower C-W wetting transition temperature is attributed to higher pillar, which produces more work against resistance to require more energy in the C-W wetting transition. Thus, the superhydrophobic surface with a higher pillar structure exhibits more stable non-wetting property at a low temperature. The presented work is not restricted to understanding the regime behind low-temperature non-wetting behaviors, but exhibits advances in theoretical research.

**Supplementary Materials:** The following are available online at https://www.mdpi.com/article/10.3390/coatings11050522/s1. Figure S1: XPS high-revolution spectra of substrate for O 1s, C 1s, F 1s, and Si 2p, Figure S2: Schematic diagram of the experiment setup, Figure S3: The real-time surface temperature under cooling conditions.

**Author Contributions:** Conceptualization, Y.S.; methodology, X.X.; validation, X.X., H.C., Z.C., S.L. and J.J.; writing—original draft preparation, X.X.; writing—review and editing, Y.S.; visualization, X.X. and J.J.; supervision: J.T.; funding acquisition, Y.S. and J.T. All authors have read and agreed to the published version of the manuscript.

**Funding:** This work was supported by the National Natural Science Foundation of China (No. 52075246, 12002364), and the NUAA Innovation Program for Graduate Education (kfjj20200613, kfjj20200605).

**Institutional Review Board Statement:** Not applicable.

**Informed Consent Statement:** Not applicable.

**Data Availability Statement:** The data presented in this study are available within the article.

**Conflicts of Interest:** The authors declare no conflict of interest.

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
