# Peer review of "Mechanical Equilibrium Dynamics Controlling Wetting State Transition at Low-Temperature Superhydrophobic Array-Microstructure Surfaces"

_coatings, doi:10.3390/coatings11050522_

Round 1

Reviewer 1 Report

Contents of the research and experimental procedure seems to be of high quality, and the reviewer does not have much objection against the publication of this paper. 

Some suggestions on the formatting nature;

  1. Consider adding a small section within the introduction section explaining the Cassie Baxter principles, it would make it more friendly for readers who are not familiar with the concept.
  2. There are two section 4s (conclusion should be changed to 5. Conclusions) 
  3. Parts of Section 3.1, lines 97 to 103 (?) should perhaps be considered to be oved to Section 2, and elaborate more in details about the specimen assembly procedure. A significant importance of this experimental regime is based on the pore sizes of the substrate (the superhydrophobic surface), and more details should be included for readers who may wish to replicate this experiment. 

Author Response

Response to Reviewer

Thank you very much for your positive evaluations and the valuable suggestions on our manuscript. We have revised carefully the paper and addressed all these comments. Also, the modifications and supplements were clearly highlighted using yellow in revised manuscript. The main corrections and the response to reviewer’s comments are listed as follows:

I look forward to hearing from you soon.

Best regards!

Assoc. Prof. Yizhou Shen

Nanjing University of Aeronautics and Astronautics, P. R. China

Contents of the research and experimental procedure seems to be of high quality, and the reviewer does not have much objection against the publication of this paper.

Some suggestions on the formatting nature;

  1. Consider adding a small section within the introduction section explaining the Cassie Baxter principles, it would make it more friendly for readers who are not familiar with the concept.

Thanks very much! According to your suggestion, we added the explanation of Cassie-Baxter principle at lines 40-42 in introduction section.

  1. There are two section 4s (conclusion should be changed to 5. Conclusions)

We updated the conclusion to 5. Conclusions.

  1. Parts of Section 3.1, lines 97 to 103 (?) should perhaps be considered to be oved to Section 2, and elaborate more in details about the specimen assembly procedure. A significant importance of this experimental regime is based on the pore sizes of the substrate (the superhydrophobic surface), and more details should be included for readers who may wish to replicate this experiment.

Thank you for your suggestion. We adjusted Section 3.1 to Section 2 and modified some content to make this section more logical, as shown in lines 98-110. Also, we added a description of the acquisition of diameter and height for pillar array structure in lines 87-88. The preparation of hydrophobic SiO2 nanoparticles was provided in detail in Supplementary Material.

Reviewer 2 Report

The authors present in paper: Mechanical equilibrium dynamics controlling wetting state transition at low-temperature superhydrophobic array-micro- structure surfaces some very interesting results on the contact behavior of liquids with a metallic surface for possible applications at industrial scale.

1. The major concern are the references to Supplementary Material (SM) for section 1,2,3,4 and 5 (lines (L):88, 125, 147, 159, 260, 290-293 etc.) but it is not possible to analyze the S M – is not provided by the authors so is difficult to connect the mentions from the paper with no results – please provide a link to SM or upload on the platform the SM file.

2. L17: re-phrase the sentence Here, we found ……

3. L42: delete , after temperature and use a . after etc.

4. Why do you use Ti6Al4V as substrate: any particular reason? For applications?

5. L80-81: chemical composition is a producer report? give a reference

6. L82: mention the period of ultrasound cleaning, 30, 60 or more minutes

7. L262 – L267 relationships 1 and 2 are probably more complex at micro scale and the approximation is quite forced, please reconsider and re-phrase the comments about these equations.

Author Response

Response to Reviewer

Thank you very much for your positive evaluations and the valuable suggestions on our manuscript. We have revised carefully the paper and addressed all these comments. Also, the modifications and supplements were clearly highlighted using green in revised manuscript. The main corrections and the response to reviewer’s comments are listed as follows:

I look forward to hearing from you soon.

Best regards!

Assoc. Prof. Yizhou Shen

Nanjing University of Aeronautics and Astronautics, P. R. China

 The authors present in paper: Mechanical equilibrium dynamics controlling wetting state transition at low-temperature superhydrophobic array-micro- structure surfaces some very interesting results on the contact behavior of liquids with a metallic surface for possible applications at industrial scale.

  1. The major concern are the references to Supplementary Material (SM) for section 1,2,3,4 and 5 (lines (L):88, 125, 147, 159, 260, 290-293 etc.) but it is not possible to analyze the S M – is not provided by the authors so is difficult to connect the mentions from the paper with no results – please provide a link to SM or upload on the platform the SM file.

Thanks very much! According to your suggestion, we uploaded the Supplementary Material to the platform, and the platform would provide a download link.

  1. L17: re-phrase the sentence Here, we found ……

 We re-edited this sentence in lines 17-19, and checked the English writing.

  1. L42: delete , after temperature and use a . after etc.

 Thank you for your suggestion. We corrected the mistake.

  1. Why do you use Ti6Al4V as substrate: any particular reason? For applications?

The purpose of our work is to guide the application of superhydrophobic materials in the field of anti-icing for aircraft surface. However, most of the aircraft shells are made of Titanium alloy. According to the practical application, it is appropriate to use Ti6Al4V in the experiment.

  1. L80-81: chemical composition is a producer report? give a reference

The chemical composition of Ti6Al4V was mentioned in our previous publication. We gave the reference as shown in https://aip.scitation.org/doi/suppl/10.1063/1.4984230.

  1. L82: mention the period of ultrasound cleaning, 30, 60 or more minutes

Thanks for your kindly reminding. The period of ultrasound cleaning was 30 minutes and had been added to the line 85 in manuscript.

  1. L262 – L267 relationships 1 and 2 are probably more complex at micro scale and the approximation is quite forced, please reconsider and re-phrase the comments about these equations.

In our discussion, these relationships were described qualitatively. Actually, according to the experimental data, relationships 1 and 2 were valid. However, considering that the experimental data had a certain error, we choose the approximation to better ensure the correctness of the relationship. The detailed process of relationship 1 was provided in the Supplementary Material, and was extracted as follows:

According to the analysis of work done against resistance, the penetration depth on the H2 and H3 surfaces reflects the relationship of work done against resistance. The penetration results the decrease of the droplet volume on the superhydrophobic surface, the penetration depth can be easily obtained by change of droplet volume. We assume that penetration is an ideal state, the area of liquid-air interface is constant, which is expressed as . The volume of a droplet on the superhydrophobic surface is ,  is the radius of a 10-μL droplet. Therefore, the penetration depth is ,  and  are the contact angle (CA) for the initial Cassie-Baxter wetting state and C-W wetting transition point. Here, the  on H2 and H3 surfaces are 150.4° and 150.7°, and the  are 144.5° and 140.3°, respectively. According to the CA, the penetration depth on the H2 and H3 surfaces can be obtained, which are expressed as  and  respectively. The relationship of the penetration depth on the H2 and H3 surfaces is . The work done against resistance on the H2 and H3 surfaces are expressed as  and , the relationship is .

Round 2

Reviewer 2 Report

I accept the paper in its current form